# Navigation Data Anomaly Analysis and Detection

**Ahmed Amro \***, **Aybars Oruc**, **Vasileios Gkioulos and Sokratis Katsikas**

Department of Information Security and Communication Technology, Faculty of Information Technology and Electrical Engineering, Norwegian University of Science and Technology, 2815 Gjøvik, Norway; aybars.oruc@ntnu.no (A.O.); vasileios.gkioulos@ntnu.no (V.G.); sokratis.katsikas@ntnu.no (S.K.)
\* Correspondence: ahmed.amro@ntnu.no

**Abstract:** Several disruptive attacks against companies in the maritime industry have led experts to consider the increased risk imposed by cyber threats as a major obstacle to undergoing digitization. The industry is heading toward increased automation and connectivity, leading to reduced human involvement in the different navigational functions and increased reliance on sensor data and software for more autonomous modes of operations. To meet the objectives of increased automation under the threat of cyber attacks, the different software modules that are expected to be involved in different navigational functions need to be prepared to detect such attacks utilizing suitable detection techniques. Therefore, we propose a systematic approach for analyzing the navigational NMEA messages carrying the data of the different sensors, their possible anomalies, malicious causes of such anomalies as well as the appropriate detection algorithms. The proposed approach is evaluated through two use cases, traditional Integrated Navigation System (INS) and Autonomous Passenger Ship (APS). The results reflect the utility of specification and frequency-based detection in detecting the identified anomalies with high confidence. Furthermore, the analysis is found to facilitate the communication of threats through indicating the possible impact of the identified anomalies against the navigational operations. Moreover, we have developed a testing environment that facilitates conducting the analysis. The environment includes a developed tool, NMEA-Manipulator that enables the invocation of the identified anomalies through a group of cyber attacks on sensor data. Our work paves the way for future work in the analysis of NMEA anomalies toward the development of an NMEA intrusion detection system.

**Keywords:** NMEA; cybersecurity; anomaly analysis and detection; maritime

## 1. Introduction

The maritime domain is undergoing a major digital transformation, leading to substantial changes in the business models, processes, and technology [1]. The Integrated Navigation System (INS) on conventional vessels of today is deployed to support safe navigation as a result of such digital transformation. However, technological advancements would change the characteristic of vessels dramatically in the near future. Recently, new projects have been proposed to increase autonomy in maritime. This includes automating maritime systems and services until such systems can reach sea-going autonomous operation by the year 2035 [2]. These projects have led to the proposition of a new ship class named Maritime Autonomous Surface Ship (MASS) as defined by the International Maritime Organization (IMO) [3]. Among these new projects is the Autonomous Ferry (Autoferry) project [4]. The project aims to develop an Autonomous Passenger Ship (APS) or ferry for carrying passengers across the Trondheim city canal in Norway. The APS is expected to be remotely monitored and controlled when necessary from a remote center.

The novelty of the Autoferry project influenced the cyber risk paradigm and led to unique attack objectives and techniques. The main factors that have led to this are the auto-remote operational mode as well as the fact that passengers will be on board without a crew. The auto-remote operational mode has led to novel and broader cyber attack vectors due to

remote connectivity, dependency on automated services, reduced human defenses, and the dependency on digital technologies for the remote operator for intervention. Furthermore, the presence of passengers imposes a safety risk factor motivating different types of threat actors to cause them harm through cyber attacks. Among the identified attack vectors in Autoferry is the navigational information which is communicated among the different marine components.

The National Marine Electronics Association (NMEA) defined a group of electronic and data specifications for the communication between different marine electronic systems. These specifications have manifested into a series of standards. The latest versions are NMEA0183 [5] and NMEA2000 [6]. These standards govern the structure and the manner in which messages are communicated among the different devices. NMEA messages are mainly utilized in the maritime domain. However, the positioning information provided by them has found their application in other domains such as those with requirements for location tracking for personal security [7,8], and car theft detection [9]. While these messages provide an abundance of information utilized in different navigational tasks and functions, their security has been investigated and found to be lacking any controls such as authentication, encryption, and validation [10]. This makes them susceptible to a wide range of cyber-attacks.

This paper aims to improve the security of NMEA messages by identifying and proposing relevant approaches for the treatment and monitoring of the risks associated with them. The NMEA0183 standard is considered in this paper, with future plans to extend the work to include the NMEA2000 standard. Therefore, we propose a systematic approach for analyzing NMEA messages, their anomalies, malicious causes of such anomalies (i.e., attacks) as well as the appropriate detection algorithms.

We utilize two maritime use cases throughout this paper to facilitate the description of our approach. The use cases are the APS and conventional vessels equipped with an INS. In this way, we caught an opportunity to prove the importance of our study not only for today's vessels but also for potential vessels of the future. We argue that our approach can aid in the development of resilient navigation systems that are developed and operated under the consideration of adversarial behavior.

The contribution of the paper is as follows:

- We propose a novel systematic approach for anomaly detection in NMEA messages.
- We present an analysis of possible anomalies in NMEA messages and their cause-and-effect relationship with a range of cyber-attacks.
- We propose a method for creating synthetic datasets with both normal and maliciously tampered with NMEA messages, and we implement and use a software package to create such experimental datasets.
- We use the datasets within the context of two use cases to evaluate the performance of anomaly detection approaches specifically designed for the purpose.

The remainder of the paper is structured as follows: In Section 2, we provide the necessary background and we review the relevant literature. In Section 3, we present our proposed method for systematic, multidimensional analysis of anomalies in NMEA messages. In Section 4, we discuss how our proposed method applies to the INS and the APS use cases. In Section 5, we present results of our experimentation with the proposed approaches, towards assessing its usefulness in developing an intrusion detection system for NMEA messages. In Section 6, we evaluate and discuss the results and findings of the experimentation, and in Section 7 we present deployment options of an NMEA IDS. Finally, Section 8 summarizes our conclusions and proposes directions for future research.

## 2. Background and Related Work

Several publications which point out cyber risks of autonomous ships are available in the literature. Kavallieratos et al. [11] presented the results of cyber risks assessment of remotely controlled and autonomous ships. The risk assessment was performed using the STRIDE threat modeling methodology. According to the results, the Automatic Identifica-

tion System (AIS), Electronic Chart Display and Information System (ECDIS), and Global Maritime Distress Safety System (GMDSS), in particular, include various high risks. Vinnem and Utne [12] discussed the possibility of using autonomous ships for damaging the offshore industry. A cyber attack may cause the collision of an autonomous ship and an offshore platform at sea, intentionally or unintentionally. The paper also suggests several mitigation measures. Keeping a small number of crew onboard is argued to be the most effective preventive measure against cyber risk according to the authors.

Not only autonomous ships but also conventional vessels sailing at sea today could be exposed to cyber attacks. Svilicic et al. [13] unveiled the cyber vulnerabilities of an INS onboard ship. The authors acquired a total of 27 pieces of information, and four vulnerabilities using a vulnerability scanner. One of the detected vulnerabilities in the INS was reported as "Critical". Moreover, a survey of intrusion detection in vehicles, including maritime vessels, is presented by Loukas et al. [14]. The authors discussed several works targeting Global Positioning System (GPS) and AIS spoofing and manipulation. However, no reference is made to the NMEA protocol.

Motivated by the identified threats in the maritime industry and focusing on the NMEA protocol as a possible threat vector, we surveyed the current state-of-the-art of NMEA security. Krile et al. [15] explained the network of an INS onboard ship, including a detailed description of the NMEA 0183 and 2000 standards. The authors focus on NMEA 2000 in particular in different aspects, such as components of an NMEA 2000 network, comparison of ethernet and Controller Area Network (CAN), and the functions of CAN. Moreover, the authors discuss NMEA software, including Sail Soft NMEA Studio, Maretron N2K Analyzer, and N2K Meter. Additionally, several applications of NMEA messages have been observed in digital forensics [16,17], personal security [7,8], car theft detection [9], as well as utilizing NMEA messages in detection Global Navigation Satellite System (GNSS) Spoofing [18]. Nevertheless, these works did not discuss the security of NMEA messages themselves. Some works have argued that NMEA security currently depends on the network and host security [19,20]. However, Seong and Kim [21] addressed the cybersecurity of NMEA messages by utilizing secure hash functions when storing NMEA messages in the voyage data recorder onboard vessels. This is argued to improve the authenticity of stored NMEA messages.

Additionally, Boudehenn et al. [22] proposed a machine learning approach to detect GPS attacks. The GPS device broadcasts NMEA 0183 messages to the ship network. Machine learning software developed by the authors in a Raspberry Pi 3B+ could detect GPS jamming and spoofing attacks successfully. In this way, the officer of the watch on the bridge may be notified about a potential GPS attack. Machine learning could be also used to detect malicious activities in the ship network [23]. Moreover, Hemminghaus et al. [24] have presented a bridge attack tool named BRidge Attack Tool (BRAT) that targets NMEA messages with a wide range of attacks in order to assess the security of maritime systems. The authors discussed the lack of security in marine systems, particularly the ones utilizing the NMEA protocol. Then, they presented the architecture of the tools and evaluated it against the open-source OpenCPN chart plotter.

Another application of the NMEA protocol is found in AIS. The vessels are equipped with an AIS to improve the safety and efficiency of navigation and to protect the marine environment [25]. It is a compulsory component for vessels under specific conditions described in the Safety of Life at Sea (SOLAS) Convention [26]. An AIS tranceiver transmits static, dynamic, and voyage related information as well as safety related messages using the format of NMEA messages [25,27]. Several works have addressed anomaly detection in AIS. Iphar et al. [28] proposed an integrity assessment of AIS messages from a data quality perspective. The authors targeted AIS messages for data quality assessment and conducted several manipulation functions on AIS messages to invoke anomalies in the data. Then they proposed a rule-based detection approach. In another work, Blauwkamp et al. [29] utilized machine learning for detecting anomalies in traffic inferred from AIS messages. Although the authors did not target cybersecurity, cyber attacks are among the main

motivations of their research. Although NMEA and AIS messages have a relatively similar format, AIS messages include encoded binary payload instead of a textual payload in the NMEA-0183. Furthermore, AIS messages carry different information than NMEA messages, such as traffic messages from other ships. These differences motivated the work in this paper to investigate suitable anomaly analysis and detection approaches.

The origin of NMEA comes from the CAN protocol or CAN bus, a message-based protocol that enables communication among devices in automobiles [30]. Several works in the literature have addressed anomaly detection in CAN bus. We aim to infer relevant artifacts from the domain of CAN bus anomaly detection and utilize them for NMEA anomaly detection. In this paper, we rely on the state-of-the-art Intrusion Detection Systems (IDS) for CAN bus in the automotive domain which was captured by Lokman et al. [31]. The authors discussed several aspects, namely deployment strategies, detection approaches, attacking techniques, and technical challenges related to the field. Due to the similarities between NMEA and CAN bus, several artifacts were found relevant to our work and they will be discussed throughout this paper.

The systematic anomaly analysis of NMEA messages proposed in this paper is influenced by the Six-Step Model proposed by Sabaliauskaite et al. [32]. The authors proposed six steps for conducting joint safety and security risk analysis process utilizing six dimensions, namely, functions, structure, failures, attacks, safety countermeasures, and security countermeasures. Accordingly, the anomaly analysis in this paper consists of six steps; each step analyzes a different dimension related to NMEA anomaly detection, namely, navigational functions, messages, fields, anomalies, attacks, and detection methods. The systematic and multidimensional nature of the analysis in the Six-Step model has influenced our proposition. Additionally, the analysis of NMEA messages and their anomalies is influenced by the AIS analysis process conducted by Iphar et al. [28] (more details in Section 4.4).

The attack procedures in our work are influenced by the domain-specific information provided in the work of Hareide et al. [33]. The authors in [33] have discussed a cyber kill chain in maritime toward increasing the navigators' preparedness against cyber attacks. Specifically, they have conducted a contextual attack targeting navigational information received at the ECDIS. Moreover, we relied on the *ATT&CK* framework [34] to describe the conducted attack techniques in the attack scenarios. The *ATT&CK* framework was chosen due to its comprehensive threat model in describing adversarial behavior.

## 3. Methodology

This paper focuses on the detection of anomalies in NMEA messages that can be caused by malicious actors. Figure 1 depicts the proposed meta-model of the NMEA anomaly detection system. Several NMEA message types support several navigational functions. Each message consists of several fields, each holding specific information. Attackers conduct attack procedures to impact navigational functions by targeting message types or fields. Defenders implement detection algorithms to protect the navigational functions by monitoring message types and fields to detect attack procedures. The meta-model is general in nature; as such, it is relevant to any use case that utilizes sensor data communicated in NMEA massages for navigational functions.

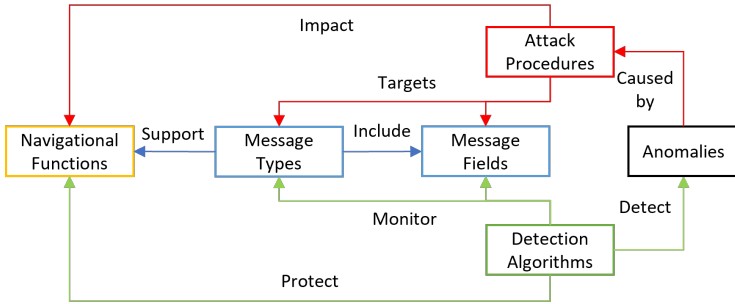

**Figure 1.** NMEA anomaly detection meta model.

We propose a method for systematic and multidimensional anomaly analysis of NMEA messages toward the development of an NMEA-focused anomaly detection solution. Anomaly analysis, as defined in the data quality domain, is a process for analyzing the values in a data set empirically, looking for unexpected behavior [35]. In this process, NMEA messages are analyzed for identifying possible anomalies, their impact, and ways in which they can be invoked and detected. A detailed description of the proposed method is provided hereafter:

### 3.1. Step 1—Navigational Functions (i.e., Tasks)

The identification of the navigational functions which rely on NMEA messages. These functions are defined for an INS by the IMO and for autonomous ships by classification societies describing the different tasks to be carried by marine systems or personnel such as route monitoring, collision avoidance, engine monitoring and control, and others. This information can later be utilized in risk analysis and, specifically, impact assessment.

### 3.2. Step 2—Message Types

The identification of targeted message types, and categorizing them according to relevant attributes such as their navigational functions (e.g., engine monitoring) and source (e.g., engine). This step specifies the scope of the analyzed messages and is expected to be system-dependent, since each system supports a specific list of messages.

### 3.3. Step 3—Message Fields

The identification of relevant message fields, the identification of the type of information they hold (e.g., speed, heading, etc.), and the format of each field. The type of information is useful for the identification of related message fields within the same message and across different message types. The information is useful for both attack and detection activities. A sophisticated attacker is expected to reflect a completely modified view, while the defender can detect anomalies by observing relevant fields for inconsistencies. On the other hand, the format and other aspects such as the range of each field are useful for the development of anomaly detection methods.

Steps 2 and 3 rely heavily on the format of the analyzed messages. The format of an NMEA-0183 message is depicted in Figure 2. After the starting delimiter, ($), a 2-letter NMEA talker ID (e.g., GP for GPS) is attached to a 3-letter message ID specifying the message type (e.g., DTM, RMC, etc.). Then, each NMEA message has several fields, each corresponding to a certain piece of information, such as time, longitude, Speed Over Ground (SOG) etc. Then, a 2-digit in hexadecimal format that represents the calculated sentence checksum is separated from the last field value using the checksum delimiter ($*$). Finally, a carriage return and a line field specify the end of each message.

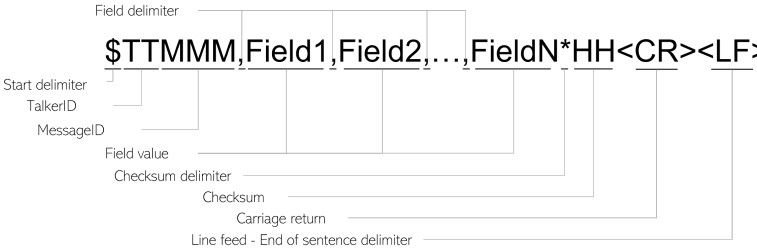

**Figure 2.** The format of an NMEA message.

### 3.4. Step 4—Anomalies

The identification of anomalous patterns (e.g., unusual values and events) that may appear during operations. During this step, all messages within the scope and their fields are analyzed to identify anomalous patterns based on some categorization of anomalies.

*3.5. Step 5—Attack Techniques*

The identification of attack techniques that can be carried out to invoke anomalous patterns in the selected message types and their message fields.

*3.6. Step 6—Detection Algorithms*

The identification of suitable detection algorithms for detecting anomalous patterns caused by attack techniques carried against NMEA messages. This step is tightly coupled with the previous step as the detection algorithm is continuously challenged and enhanced with improved attack techniques until a sufficient efficiency level is achieved.

## 4. Systematic NMEA Analysis Considering APS and INS Use Cases

In this section, we discuss the activities, artifacts, and results of our proposed NMEA analysis approach presented in Section 3, considering both the INS and the APS use cases. This is aimed to demonstrate the utility of the proposed anomaly analysis process in addition to the development of a suitable anomaly detection solution.

*4.1. Step 1—Navigational Tasks and Functions*

The tasks and functions for the INS and APS use cases were identified. The tasks of the INS were defined in the Resolution MSC.252(83) "Adoption of the revised performance standards for Integrated Navigation System (INS)" by the IMO [36]. On the other hand, the functions for the APS are defined by Amro et al. [37].

4.1.1. Navigational Tasks of the INS

The concept of the INS was developed to enhance the safe navigation of vessels with integrated and augmented functions. The INS consists of six navigational tasks [36], as follows:

- Route Monitoring (INS-RM): continuous monitoring of the own vessel as per the planned route [38].
- Route planning (INS-RP): capability of route planning (e.g., store and load, import, export, documentation), route checking based on minimum under keel clearance, drafting and refining the route plan against meteorological information [36].
- Collision Avoidance (INS-CA): detecting and plotting other ships and objects in the vicinity in order to prevent collisions [38].
- Navigation Control Data (INS-NCD): providing data to the task station for the manual and automatic control of the ship [38].
- Navigational Status and Data Display (INS-NSDD): displaying several information (e.g., AIS data, Maritime Safety Information (MSI) messages, INS configuration), and providing management functions [36].
- Alert management (INS-AM): centralized alert management on the bridge for the monitoring, handling, distribution, and presentation [38].

The INS facilitates the performing of the aforementioned navigational tasks. The tasks of "route monitoring" and "collision avoidance" are mandatory as per the IMO's regulations [38]. Moreover, the requirements of "presentation of navigation control data for manual control" of the navigation control data task and "Module C" of the alert management task should be fulfilled [36]. Given that the lack of some navigational tasks and requirements in the INS may increase risks in the safe navigation of the vessel, they are classified as mandatory by the IMO. For instance, while "collision avoidance" and "route monitoring" are mandatory navigational tasks for an INS, the "route planning" and "navigational status and data display" are left optional by the IMO [36]. In the next step, the relevant NMEA messages to each navigational task is identified. Such a matching enables us to understand the risk level of potential NMEA anomalies by considering mandatory and optional navigational tasks defined by the IMO.

### 4.1.2. The Functions of the APS

There exists no regulatory framework or globally accepted guidelines that define the functions of an APS. Nevertheless, in our previous work [37], we have compiled a group of expected APS functions based on a group of relevant works including the work of Rødseth et al. [39] in the "Maritime Unmanned Navigation through Intelligence in Networks (MUNIN)" project as well as class guidelines for autonomous and remotely operated vessels by DNV [40]. A brief summary of the expected APS functions is discussed below (refer to [37] for more details):

- Engine Monitoring and Control functions: the monitoring and control of APS engine. They can be conducted by the APS itself (APS-AEMC), a Remote Control Center (RCC) (APS-REMC), or an Emergency Control Team (ECT) (APS-EEMC).
- Navigation Functions: establishing situational awareness. They can be conducted by the APS itself based on the sensor data (APS-AN), at the RCC based on the sensor data transmitted from the APS (APS-RN), or by the ECT based on the sensor data transmitted from the APS (APS-EN).

Impacting these functions through cyber attacks could cause safety, financial, and operational consequences according to a previously conducted risk assessment [41].

### 4.2. Step 2—Message Types

There are many NMEA messages (i.e., sentences) defined in the IEC 61162-1 standard [42]. In this paper, we will restrict our analysis on the NMEA messages broadcasted by the Bridge Command simulator (https://www.bridgecommand.co.uk/ (accessed on 16 February 2022)) in order to evaluate the proposed analysis process. The messages were investigated using the guideline of the IEC 61162-1 standard [42] which is compatible with NMEA 0183. The messages in addition to their descriptions are shown in Table 1.

**Table 1.** NMEA messages within the analysis scope.

| Msg. | Description | Msg. | Description |
|------|-------------|------|------------|
| DTM | Datum reference | GGA | Global positioning system (GPS) fix data |
| GLL | Geographic position—Latitude/longitude | HDT | Heading true |
| RMC | Recommended minimum specific GNSS data | ROT | Rate of turn |
| RPM | Revolutions per minute | RSA | Rudder sensor angle |
| TTM | Tracked target message | ZDA | Time and date |

After identifying the targeted NMEA messages for analysis, their involvement in the navigational functions is analyzed. This analysis can reflect the impacted navigational function for each NMEA message that is a subject of an attack. Table 2 depicts the identified relevance between messages and the APS and INS functions. A message is considered relevant to a function if it provides a piece of information that influences performing the function. For instance, the APS-AN function relies on the location information (i.e., coordinates) communicated in either one of the GGA, GLL, or RMC messages for route planning. Regarding the INS use case, the targeted NMEA messages are involved in all the INS functions except "Alert Management" using Resolution MSC.252(83) [36]. On the other hand, as the APS use case is in its the early development stages, we considered the possible involvement of each message in the APS navigational functions based on the designs and concepts communicated in the literature. Our analysis suggests that all considered messages are expected to be involved in the navigation functions except for the RPM messages, which are expected to be involved in the engine monitoring and control functions.

**Table 2.** Mapping of the messages and their supported navigational function.

| Message \ Function | INS-RM | INS-RP | INS-CA | INS-NCD | INS-NSDD | INS-AM | APS-AN | APS-RN | APS-EN | APS-AEMC | APS-REMC | APS-EEMC |
|---|---|---|---|---|---|---|---|---|---|---|---|---|
| DTM | ✓ |  |  |  | ✓ |  | ✓ | ✓ | ✓ |  |  |  |
| GGA | ✓ |  |  |  | ✓ |  | ✓ | ✓ | ✓ |  |  |  |
| GLL | ✓ |  |  |  | ✓ |  | ✓ | ✓ | ✓ |  |  |  |
| HDT | ✓ |  |  | ✓ | ✓ |  | ✓ | ✓ | ✓ |  |  |  |
| RMC | ✓ |  |  | ✓ | ✓ |  | ✓ | ✓ | ✓ |  |  |  |
| ROT | ✓ | ✓ |  | ✓ | ✓ |  | ✓ | ✓ | ✓ |  |  |  |
| RPM |  |  |  | ✓ | ✓ |  |  |  |  | ✓ | ✓ | ✓ |
| RSA |  |  |  | ✓ | ✓ |  | ✓ | ✓ | ✓ |  |  |  |
| TTM | ✓ |  | ✓ |  |  |  | ✓ | ✓ | ✓ |  |  |  |
| ZDA |  |  | ✓ | ✓ | ✓ |  | ✓ | ✓ | ✓ |  |  |  |

*4.3. Step 3—Message Fields*

During this step of our analysis, we have analyzed the fields of all the messages identified during step 2 (Section 4.2). The goal of this analysis is to understand the utility, and format of the piece of information depicted in each field. This understanding facilitates the activities to be conducted in the upcoming steps. Furthermore, the related NMEA messages are identified and depicted in Table A1 in Appendix B. Two messages are considered to be correlated if a change in information contained in one message that occurs under normal circumstances, will (direct effect) or might (indirect effect) change information in the other message. An example of an indirect effect can be observed in the relation between the RMC and RPM messages: changes in the Speed over Ground (SOG) in the RMC message might reflect different engine speed which is captured in the revolutions per minute field within the RPM message. On the other hand, an example of a direct effect has been observed in the location information (i.e., longitude and latitude) that is communicated in three messages, namely, GGA, GLL, and RMC. If any value changes in any message, it should be reflected in the other messages. Such information is valuable for both attack and detection activities.

*4.4. Step 4—Anomalous Patterns*

In this step, the possible anomalies that can be observed in NMEA messages are identified. Iphar et al. [28] proposed 13 possible anomalous patterns in AIS messages, that follow the same standard as NMEA with some differences in the message format as well as in content. However, they have the same abstraction of message types, each consisting of several message fields. In this paper, we adopt the relevant anomalies and neglect those that are not relevant to NMEA messages. Seven main anomalous patterns have been identified; these, along with brief descriptions are shown in Table 3.

**Table 3.** The anomalies within scope.

| Anomaly | Description |
|---|---|
| Sudden unexpected change | an abnormal change in a field value in a certain period of time. |
| Nonexistent value | a value that does not match the system specification. |
| Unexpected value | a field value that is outside the usual norm. |
| Incorrect value | a value that does not match a reference value (e.g., time). |
| Data field evolution | an abnormal pattern in a field value over time. |
| Conformity issues | values that are not within the protocol specifications. |
| Unusual reporting | reduced or increased reporting rate over a period of time |

We have analyzed all the aforementioned anomalies against all message types and their corresponding fields and recorded our results for the next steps. Some examples of the identified anomalies are presented in Table 4 while a list of all the identified anomalies is provided in Table A2 in Appendix C.

**Table 4.** Examples of possible NMEA anomalies.

| Anomaly | Message | Field | Description |
|---|---|---|---|
| Sudden unexpected change | RMC | SOG | The Speed over Ground (SOG) has abnormally changed. |
| Nonexistent value | TTM | Target Distance | Target distance larger than radar range |
| Unexpected value | ROT | Rate Of Turn | Abnormal rate of turn value |
| Incorrect value | RMC | UTC | The UTC timestamp is not correct compared to a reference time value |
| Data field evolution | TTM | Target Status | Abnormal patter in the target status over time |
| Conformity issue | RPM | Source | The source field contains values that are not either E (Engine) or S (Shaft). |
| Under Reporting | RMC | - | The rate of receiving RMC messages is less than usual |
| Over Reporting | DTM | - | The rate of receiving DTM messages is more than usual |

### 4.5. Step 5—Attack Techniques

In this section, we discuss the activities performed during the fifth step in the analysis concerning attack techniques that are expected to invoke one or several of the anomalies identified during step 4. In this direction, we propose the application of the *ATT&CK* framework [34] for threat modeling due to its comprehensive nature and suitable level of abstraction [41]. However, other threat modeling methods can still be applied if they propose attack techniques that can be technically achieved. The threat modeling approach considers both simple attacks as well as sophisticated attacks. Lokman et al. [31] discussed several attack types against the CAN bus, namely, packet insertion, erasure, reply, and payload modification. In the *ATT&CK* framework, insertion, reply, and payload modification may fall under the Manipulation of View (MoV) attack technique [43] while packet erasure may fall under Denial of View (DoV) attack technique [44]. A brief description of each technique is provided below:

- DoV attacks entail denying the seafarers or the depending systems the ability to render a live perception of the physical environment. This is achieved by dropping one or several NMEA messages to hinder the relevant navigational functions.
- MoV attacks entail the modification of the live perception of the physical environment. This can be done in several ways:
  - Fixed: the attacker modifies the values in original NMEA messages to specific fixed values. For example, no matter what is the real speed reflects another fixed speed value. This emulates a simple threat actor using simple Man-in-the-Middle (MitM) attack rules (i.e., filters).
  - Context attacks: the attacker manipulates the messages based on the values observed in the original messages to create a gradual change. This emulates a more advanced threat actor using more sophisticated MitM attack rules. Avoiding detection is among the attacker's objectives.
  - Confusion attacks: the attacker sends crafted or repeated messages in addition to the original messages.
  - Replay attacks: the attacker replays a fixed set of messages instead of the original stream of messages.

We have analyzed the anomalies and the possible attacks that can invoke them. A mapping between an anomaly and an attack is identified if the attack, based on its definition, may result in invoking the anomaly. The identified relations are depicted in Table 5. The table can be read as follows: a DoV attack is expected to only invoke an "Under Reporting"

anomaly, while a confusion MoV attack is expected to invoke all possible anomalies except "Under Reporting".

**Table 5.** Mapping of the anomalies and the attacks that are expected to invoke them.

| Attack/Anomaly | DoV | MoV | | | |
|---|---|---|---|---|---|
| | | Fixed | Context | Confusion | Replay |
| Sudden unexpected change | | ✓ | | ✓ | ✓ |
| Nonexistent value | | ✓ | ✓ | ✓ | ✓ [1] |
| Unexpected value | | ✓ | ✓ | ✓ | ✓ [1] |
| Incorrect value | | ✓ | ✓ | ✓ | ✓ [1] |
| Data field evolution | | ✓ | ✓ | ✓ | ✓ [1] |
| Conformity issue | | ✓ | ✓ | ✓ | ✓ [1] |
| Under Reporting | ✓ | ✗ [2] | ✗ [2] | ✗ [2] | ✗ [2] |
| Over Reporting | | ✗ [2] | ✗ [2] | ✓ | ✗ [2] |

[1] If the attacker replays a fixed set of messages that are not generated by normal means (i.e., forged). [2] If the attacker maintained the normal message transmission rate.

To realize such attack techniques, we have developed a system called NMEA-Manipulator to facilitate the process of invoking the identified NMEA anomalies. NMEA-Manipulator intercepts and controls the flow of NMEA messages following a set of rules (detailed description in Section 5.1).

*4.6. Step 6—Detection Algorithms*

Lokman et al. [31] discussed several detection algorithms for detecting attacks against CAN bus messages. Three main detection approaches have been observed in the literature, signature-based, anomaly-based detection, and specification-based. A brief description of each approach is provided below in addition to our rationale for its utility in our analysis:

- Signature-based detection refers to the utilization of a specific signature or event for the detection of a specific malicious activity [45]. This would require documented attacks against NMEA messages to generate suitable signatures.
- Anomaly-based detection refers to the observing of real-time activities in a system and comparing them to normal behavior and raising an alarm when a deviation of normal behavior is observed [46]. This approach includes machine learning, frequency, statistical, and hybrid-based approaches. We argue that the machine learning and statistical approaches require a large set of data to effectively train robust models and, consequently, they are currently not viable options in our case. We have reached this conclusion after experimenting with a one-class support vector machine, and decision trees for detecting anomalies. The model evaluation has reflected poor performance mostly associated with the limited size of the data set. Since there exists no publicly available data set for the NMEA messages in the scope of our analysis, we have not pursued machine learning and statistical based approaches any further. On the other hand, frequency-based detection, considering message arrival frequency, was found relevant and is further considered for evaluation.
- Specification-based detection refers to the application of suitable thresholds and rules for describing the well-known behavior of a component [47]. We argue that this approach is the most suitable in the scope of our analysis because it does not require a large amount of data for learning. Moreover, considering the dynamic, yet predictable nature of NMEA messages, their behavior might be confined within a set of rules and thresholds (i.e., specifications). We have identified several categories of specifications, namely, physical, system, protocol, and environment specifications. A brief description of each category is provided below:
  - Physical specifications restrict the manner in which the values change over time among consecutive messages (e.g., maximum change in distance). This is related to the physical environment the NMEA messages are intended to reflect.

- System specifications restrict the values in the NMEA fields and their evolution over time for each system (e.g., maximum engine rpm, SOG acceleration, etc.). This needs to be defined for each target system.
- Protocol specifications restrict the format of the NMEA messages and their fields (e.g., UTC format in the UTC field of GGA, GGL, and RMC). This needs to be defined for each target protocol; in our analysis, NMEA0183 is utilized.
- Environment specifications restrict a range of values that are related to the operational environment. This includes time, date, longitude, latitude, datum code, and others.

We have analyzed the identified anomalies by considering the expected useful detection methods. A mapping between an anomaly and a detection method is identified if the anomaly can violate a certain specification or threshold in the corresponding detection method, based on their definitions. For instance, a sudden unexpected change in any field value, within the predefined format and range, does not violate the protocol specification of that field. Furthermore, changing the filed values alone is not expected to change the frequency of message arrival. Therefore, protocol specifications and frequency-based detection are not expected to be useful for detecting this type of anomaly. However, a sudden change in certain fields related to system, physical or environmental parameters such as speed, time, and distance, would violate the corresponding specifications. Table 6 depicts the results of our analysis. Our analysis suggests that frequency-based anomaly detection might only be suitable for under and over reporting anomalies. On the other hand, the specification-based approach can be used for the remaining anomaly types, using different specification categories.

**Table 6.** Mapping of the Anomalies and proposed detection methods.

| Anomalies | Specification-Based | | | | Anomaly-Based |
|---|---|---|---|---|---|
| | **Physical** | **System** | **Protocol** | **Environment** | **Frequency** |
| Sudden unexpected change | ✓ | ✓ | | ✓ | |
| Nonexistent value | | ✓ | | | |
| Unexpected value | ✓ | ✓ | | ✓ | |
| Incorrect value | | | | ✓ | |
| Data field evolution | ✓ | ✓ | | | |
| Conformity issue | | | ✓ | | |
| Under Reporting | | | | | ✓ |
| Over Reporting | | | | | ✓ |

In the sequel, we analyze the different detection methods against the messages and their fields to identify the required specification rules for detection.

## 5. Data Generation and Preparation

In this section, we present the results of our experiments throughout our analysis in order to evaluate its usefulness in the development of an intrusion detection solution. Initially we generated data, prepared it, and enriched it to facilitate the analysis. Afterwards we utilized the generated data for the identification of candidate specifications and rules for detecting anomalies. The activities in this process have been conducted using our developed maritime-themed cybersecurity testbed, which we proposed and presented in our earlier work [48]. The testbed includes several components that support the development of the anomaly detection solution.

### 5.1. Data Generation

The data generation process is facilitated by the availability of a simulator software or a device that generates NMEA messages. There are several NMEA simulator software, such as BridgeCommand simulator (https://www.bridgecommand.co.uk/ (accessed on 18 February 2022)) and NMEA simulator (https://github.com/panaaj/nmeasimulator

(accessed on 18 February 2022)). The BridgeCommand simulator software was utilized in this paper since it allows for customized scenarios. The customization includes the navigational area, ship class, weather, time, and options for the included technologies on board.

Additionally, the data generation process is supported by capabilities for conducting various attack scenarios in order to invoke the different analyzed anomalies. For this reason, we have developed a system called NMEA-Manipulator. Similar to the previously proposed BRAT assessment tool [24] NMEA-Manipulator enables conducting a wide range of attacks against NMEA traffic. However, the main design goal of it is not to assess the security of marine systems, rather to facilitate the analysis of NMEA anomalies toward the development of intrusion detection systems. NMEA-Manipulator is utilized in two steps, namely, step 5 of the NMEA anomaly analysis process to observe the impact of the suggested attack techniques, and it is utilized as well in the data generation step for evaluating the different detection algorithms. An overview of NMEA-Manipulator is depicted in Figure 3.

The NMEA-Manipulator must be hosted in a device that is connected to the same LAN that connects the original NMEA speaker and listener. Furthermore, the NMEA-Manipulator requires two additional Non-Developmental Items (NDI), namely, a MitM tool such as *Ettercap* and a network sniffer tool such as *tshark*. The MitM tool provides the ability to access and control the LAN traffic, while the sniffer tool allows the recording of the original NMEA messages into an NMEA Messages File (NMF). Additionally, NMEA-Manipulator requires attack rules to govern its behavior. The attack rules are inserted into an Attack Rules File (ARF). The ARF contains multiple lines, each corresponding to a certain attack scenario (i.e., invoked anomaly). The attack rules command the NMEA-Manipulator to perform one or more of the attack techniques discussed in Section 4.5.

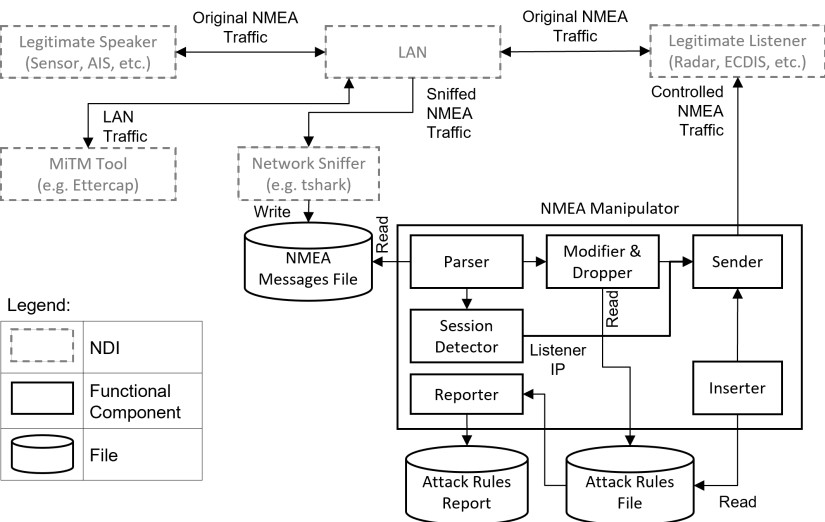

**Figure 3.** Overview of NMEA-Manipulator.

The NMEA-Manipulator includes six main subcomponents (i.e., modules), a parser, a session detector, a modifier and dropper, an inserter, a sender, and a reporter. The parser reads the NMEA messages from the NMF and invokes the modifier and dropper in case a new message is observed. The modifier and dropper then apply the activated attack rules specified in the ARF while the inserter applies the activated insertion attack rules as well as the replay and confusion attacks. The session detector identifies the IP address of the NMEA listener from the sniffed traffic and forwards it to the sender which sends the modified and inserted messages to the listener. Finally, the reporter generates a log file regarding the activated, deactivated, and modified attack rules to facilitate the later steps of the analysis.

In this direction, we utilized NMEA-Manipulator during the data generation process by conducting three experiments. Each experiment consists of several attack scenarios running in conjunction with a normal navigational scenario. The attack scenarios were chosen to be comprehensive so that they invoke a wide range of the identified anomalies. All the exercises included a normal navigational scenario which is following a predefined path in an area near the UK using a large ship equipped with a RAdio Detection Furthermore, Ranging (RADAR) and a GPS. A brief summary of the conducted experiments is presented below:

1. A combination of different MoV attacks, namely, fixed and context attacks was conducted in an attempt to invoke five anomalies, namely sudden unexpected change, nonexistent value, unexpected value, incorrect value, and data field evolution. The attack scenarios targeted several messages and message fields such as going back in time 1 day by changing UTC fields in GGA and GLL messages. Another example is increasing the distance of RADAR targets as well as other fields in the TTM message, to create a collision scenario.

2. Several MoV context attacks and DoV attacks were conducted to invoke data field evolution and under reporting anomalies, respectively.

3. A combination of different MoV attacks was conducted, namely fixed, confusion and replay attacks. The goal is to invoke several anomalies, including conformity issues and over reporting.

More details regarding the conducted experiments can be found in Table A3 in Appendix D.

The testing environment used to realize the different scenarios is depicted in Figure 4. The ship view is produced using the bridge command simulator, which is utilized as the NMEA sender. The simulator includes a simulated GPS device sending NMEA messages over UDP to the listener. On the other hand, the chart plotter view is produced by the OpenCPN chart plotter software, which is utilized as the NMEA listener. Moreover, the *Wireshark* software [49] is utilized for capturing the network traffic at the receiving node. The attacker node operates the Kali Linux operating system with the NMEA-Manipulator system.

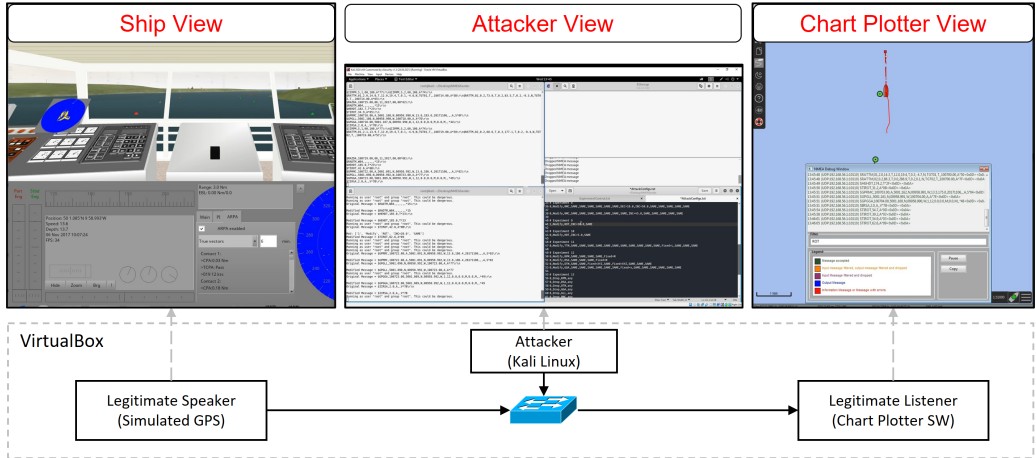

**Figure 4.** The testing environment.

Two artifacts are generated from each experiment, an experiment log and a packet capture of the traffic arriving at the NMEA receiver. The log is utilized to facilitate the labeling of NMEA messages (e.g., attacked or normal) and to support traceability of the events occurred during the experiments. The packet capture on the other hand is utilized for later steps in evaluating the different anomaly detection methods. Finally, additional experiments were conducted to capture NMEA messages in normal operations to aid the efforts in the identification of suitable specifications and rules for detecting anomalies.

In these experiments, only the NMEA sender and receiver were operational while the attacker node was kept idle.

*5.2. Preparation and Enrichment of Data*

The captured network traffic from the conducted experiments was utilized as input in this step. *Tshark*, the command-line interface of *Wireshark*, was utilized for extracting the NMEA messages with the associated time of observation of each message. The time information is needed since some NMEA messages do not contain such information. Then, data cleaning is conducted to fix some issues in the messages such as splitting concatenated massages and removing carriage return characters. Afterwards, the messages with the timing information are transformed into feature vectors. Moreover, the messages are labeled as "Normal" or "Attack" by referring to the experiment documentation. For instance, during the different experiments, some NMEA message types were not targets of attacks, which leads to a "Normal" classification of such messages and an "Attack" classification of the targeted message types during the period of activity of the attack rules. Additional fields are added for the enrichment of the captured data, such as distance from the last position, rate of message arrival etc. This is intended to facilitate the analysis of the specifications and anomalies. Finally, the output feature vectors with the headers and labels are exported into Comma-Separated Value (CSV) format. Due to different features for each NMEA message type, it has been deemed necessary to evaluate the specifications for each type separately. For this reason, the output from the previous step is a group of CSV files, one for each message type. Each file holds all the messages of a single message type that were generated in a single experiment. All activities performed during this stage except for the manual labeling of some feature vectors are automated using python scripts.

## 6. Evaluation and Discussion

We have analyzed the generated and processed data to evaluate the proposed detection approaches. A selected group of results is highlighted hereafter.

*6.1. Specification-Based Detection*

The results of the evaluation of the different proposed specification-based detection categories are presented in this section, namely, environment, physical, system, and protocol specifications.

### 6.1.1. Protocol Specifications

The approach is implemented through a validation process for each message to check the compliance of its fields with the protocol specifications. Considering the known structure and format for each message type, this approach can detect conformity issues with high confidence. For instance, an attack scenario was carried during experiment 3 (Table A3) to manipulate the view regarding the number of satellites that may be utilized for position fixing. The attack is carried by modifying the original expected integer value with the character "x". The impact of the attack was observed on the chart plotter software as the indicator of the number of satellites was changed to red, reflecting a bad signal, however, the true value was 13. A detection rule for this attack is implemented to check if the field format is in compliance or not and the results reflect correct detection.

### 6.1.2. Environment Specifications

The evaluation of this approach is not implemented, rather it is conceptually evaluated in the section. This is due to the rationale that such category of specification relies on true sources for reference information, such as correct time and datum code for coordinate ranges. The utilized simulator relies on static scenario definition files. The time of each simulation experiment is manually defined. Relying on the true system time to detect inconsistency generates false positives. Therefore, we argue that in a live implementation,

if the ship systems receive time information from a source other than GPS signals, detecting any modifications of the time information is straightforward.

### 6.1.3. System Specifications

This category of specifications is encoded in a file for each ship to host the anomaly detection solution. This file dictates to the detection software the specifications for the different systems on board to identify anomalies due to the existence of values that are not supposed to be observed in the host system.

Figure 5 depicts a visualization of the anomaly type "Non existent value" that is invoked by an MoV attack during experiment 1 (Table A3). The figure reflects two observations, the left part reflects detection of the RSA message of a sensor reading receiving from a port rudder sensor. However, the specification file of the simulated system dictates that it only has a starboard rudder sensor, which means a single rudder sensor. On the right part of the figure, a similar anomaly is detected in the RPM message when observing a new shaft (i.e., source number 4) that is not within the system specifications which is only having two shafts (i.e., source numbers 1 and 2). These two anomalies and similar ones are detected with high confidence due to the static nature of systems in ships. Furthermore, anomalies due to sudden changes can be detected using system specifications. For instance, the possible changes in the revolutions per minute (rpm) in the RPM message in one of the ship models in the simulator is in the range (25 to 2000 rpm) while the range for another model is (1 to 100). Detection rules for each one is formulated to detect any rpm changes outside the appropriate range. These rules will return anomalies with high confidence. However, if the attackers maintained a change in the rpm values that is within the system specifications, their actions will not be detected using this approach based on the rpm value as a feature. Other anomalies that can be detected using this approach are unexpected values and data field evolution. An example of an unexpected value can be observed in the rate of turn (ROT) value in the ROT message. Each ship has a certain expected limit within which the ship can turn in a minute. For instance, the difference in ROT value between two consecutive messages in one ship model in the simulator can change within the range ($-100$ to 100). An attack during experiment 2 demonstrates this effect by increasing the ROT value by a hundred while the true difference was 11.2, leading to a total difference of 111.2 which allowed the identification of the anomaly. However, similarly to the previous anomaly, an attacker can stay undetected if the resulted change is within the system specifications.

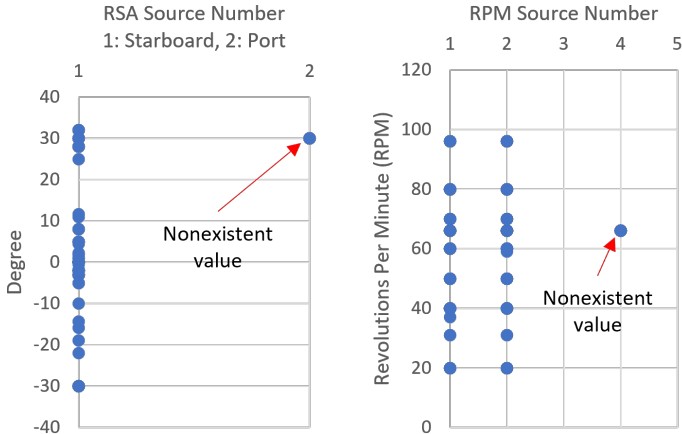

**Figure 5.** A plot visualizing the detection of a nonexistent value in a specific system.

### 6.1.4. Physical Specifications

We will demonstrate the utility and identified challenges in this approach in a selected group of messages, namely, RMC, RPM, and HDT, for detecting sudden unexpected change and data field evolution anomalies. An example of a sudden unexpected change

can be invoked through an MoV attack to reflect a position change in a strange manner. A demonstration of this was carried out during experiment 1 (Table A3). The impact of this attack can be observed clearly when visualizing the difference between each two consecutive latitude and longitude values, as depicted in Figure 6.

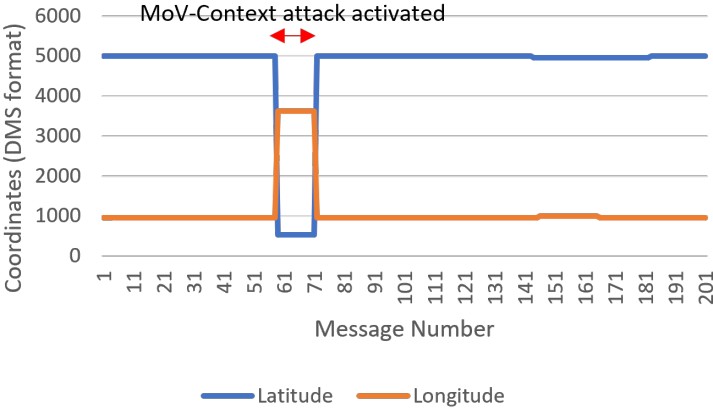

**Figure 6.** A plot visualizing the detection of a sudden change in position information in the RMC message.

A data field evolution type of anomaly can be invoked through gradually changing the SOG and Course Over Ground (COG) fields in the RMC message to reflect a different speed and heading details. An example of this attack is demonstrated during experiment 2 (Table A3). The SOG and COG values under normal conditions change in a specific manner. The SOG value between two consecutive RMC messages cannot physically change outside a certain range. Still, a visualization of a data field anomaly is depicted in Figure 7a. This anomaly can be detected by defining a rule specifying the range in which the SOG can change over a certain period of time. Then, any change outside this rule would suggest an anomaly with high confidence. Figure 7b depicts the manner in which the difference between consecutive SOG values behaves under normal and attack conditions. The impact of the attack is clearly visible and can be deduced from the figure. The behavior of the attacker can be described as reducing the SOG value by 1 several times in some time frame then returning the value to its normal value, which leads to a sudden increase. Similarly, the COG is not expected under normal conditions to have large differences between consecutive messages except if the value is reaching the limits in the range ($-360$ to $360$); only then a large change is normal. In this paper, the threshold for a normal difference was determined based on the largest calculated difference in value between the same field in two consecutive messages in the normal recorded traffic. The determination of a more accurate threshold would require a larger data set of NMEA messages from real systems.

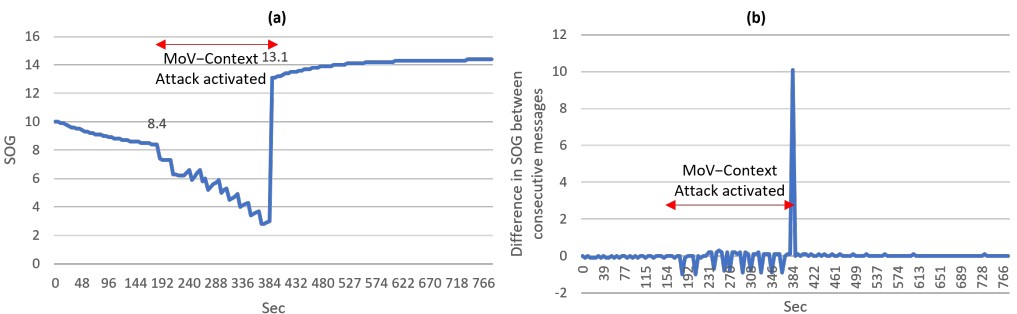

**Figure 7.** (**a**) A plot visualizing a data field anomaly in the SOG field in the RMC message (**b**) A plot visualizing the detection of a data field anomaly in the SOG field using the difference between consecutive messages.

We have mentioned previously that we are considering simple and advanced threat actors. For this reason, we have proposed the process of identifying the relevant NMEA messages and fields during step 3 (Section 3.3). If attackers crafted messages with differences within the normal range, correlating the relevant fields in other messages can be utilized to increase the confidence in the detection. A correlation between the COG field in the RMC message and the heading in the degrees field in the HDT message can provide evidence of anomalies in both messages under the condition that only one of them is subject to attack at a certain time. Figure 8a,b look very similar; this depicts the direct correlation between the two fields in two different messages. To evaluate the ability of detection based on the correlation of message fields, two attack scenarios were conducted at different times during experiment 2 (Table A3). One attack targeted the COG field by gradually increasing the COG value several times over a period of time and later returning it to normal value. The effect of this attack can be observed in Figure 8c. During the same time frame, the differences in the heading field do not reflect similar behavior; this is strong indication of an anomaly. Similarly, the other attack targeted the heading field in the HDT, causing a similar effect which is observed in Figure 8d. This also can provide strong indication of an anomaly. Notably, minor delay is sometimes observed before the COG and heading values match each other. Accommodating this constitutes a challenge for the detection algorithm and will be considered in future work.

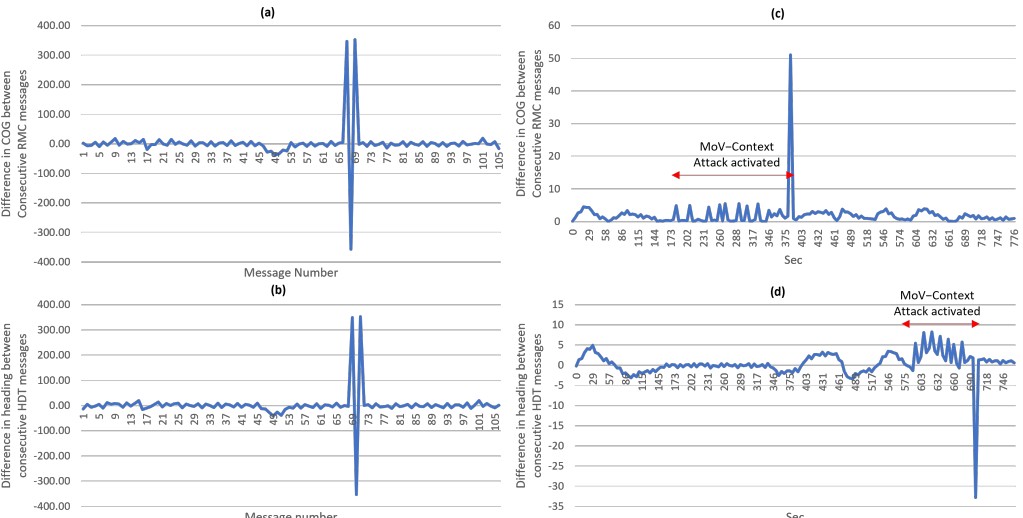

**Figure 8.** (**a**) A plot visualizing the normal manner the COG field changes between consecutive messages (**b**) A plot visualizing the normal manner in which the heading field in degrees changes between consecutive messages (**c**) A plot visualizing the detection of a data field anomaly in the COG field when correlating with the heading field in the HDT (**d**) A plot visualizing the detection of a data field anomaly in the heading field when correlating with the COG field.

### 6.2. Frequency-Based Detection

This category of specifications is solely suggested for detecting over and under reporting anomalies. The rationale behind it is that under normal conditions NMEA messages of each type have a specific amount of messages or packets arriving at the target system in a specific period. We refer to this metric as arrival rate. DoV attacks will drop some or all messages; this will lead to a decrease in the arrival rate which suggests under reporting. Furthermore, replay MoV attacks during which attackers replay messages at a reduced Transmission Rate (TR) will also cause an under reporting anomaly. On the other hand, if attackers increased the transmission rate to go beyond normal, the arrival rate will increase, which suggests over reporting. Similarly, confusion MoV can cause over or under reporting based on the transmission rate controlled by the attacker. To evaluate

the proposed detection for these anomalies, we utilize *Wireshark* to graph the number of packets per second in the recorded traffic of the relevant experiments.

The traffic contains one NMEA message per packet except for RPM messages; each pair is joined in one packet. Figure 9a,b depict the arrival rate for the recorded traffic in experiments 2 and 3 (Table A3), respectively. The impact of the DoV attacks is clearly visible in Figure 9a. During this attack, all messages were dropped, resumed, then dropped again. The third visible drop in the graph is due to the switching from the traffic generated through the NMEA-Manipulator and normal traffic. Figure 9b depicts the arrival rate during different attacks with different employed TRs. After starting with normal traffic, NMEA-Manipulator controlled the traffic and applied several fixed MoV attacks, which had a limited impact on the arrival rate. Then, a confusion MoV attack started by sending pre-recorded normal messages at TR of 1 message every 0.5 s in addition to the normal traffic. After that, the transmission rate was increased to send 1 message every 0.1 s using alternately recorded normal messages and recorded forged messages. Finally, the transmission rate was increased to 1 message every 0.05 s, similar to the last iteration, by using normal and forged recorded messages. The impact of confusion and replay attacks using different TR is clearly visible in Figure 9b as their arrival rate deviates in a clear manner from the normal arrival rate. However, if attackers targeted only a few message types, considering the arrival rate for the entire traffic might miss the targeted attack. Therefore, we propose that a dedicated arrival rate for each message type be monitored separately. This is particularly suitable in real ship systems in which different NMEA messages have different arrival rates. In order to avoid detection using frequency-based detection, attackers need to appropriately adjust the TR during their attacks, which complicates their task.

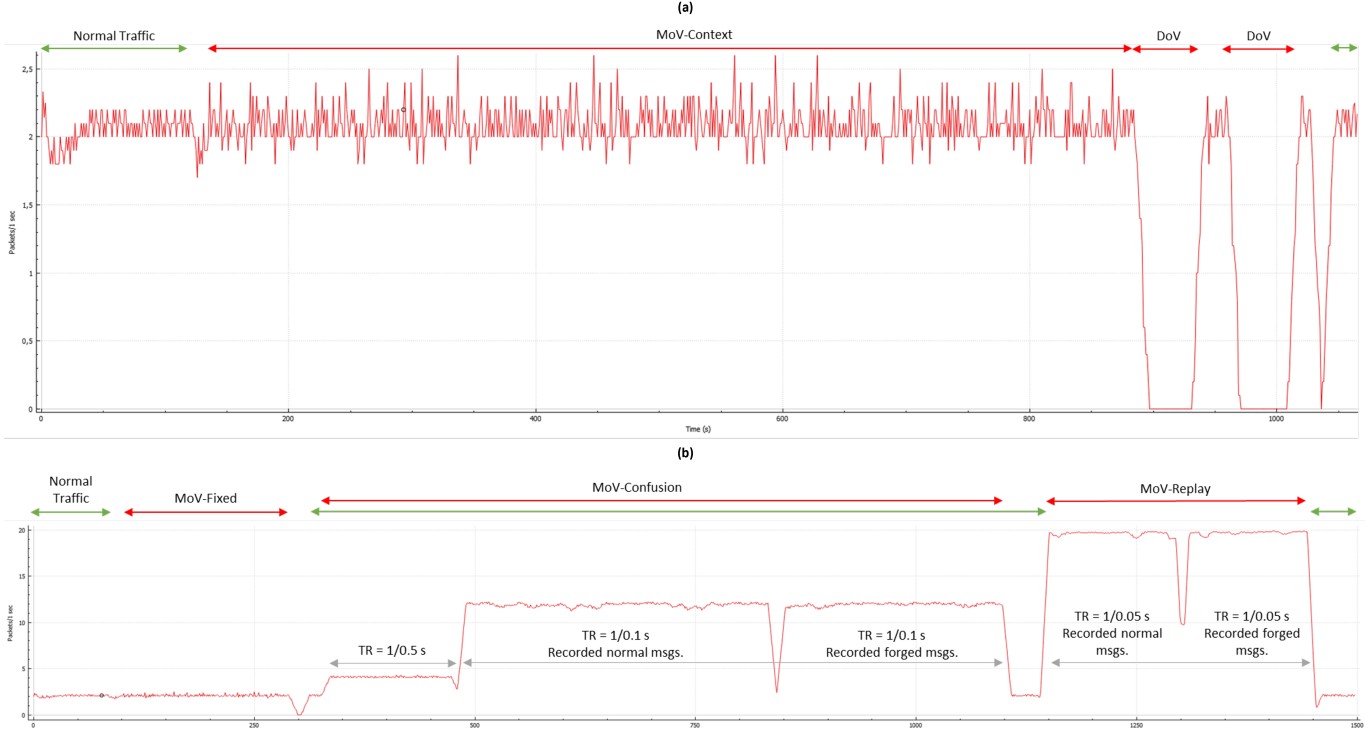

**Figure 9.** (**a**) a plot visualizing the number of packets per second during normal traffic, traffic with context MoV attacks and two DoV attacks. (**b**) A plot visualizing the number of packets per second during normal traffic, traffic with fixed MoV attacks, confusion MoV attacks, and replay MoV attacks.

### 6.3. Communication of Risk Associated with Detection

Our proposed analysis aims to develop an anomaly detection solution that is enriched with the operational context of the NMEA messages that are associated with the different functions. When an anomaly is detected using the different detection mechanisms for a

specific message or a group of messages, the impacted functions are known. Consequently, the multi-tier communication of the risk from the technical level to the operational level is facilitated. An example of this is as follows: if the anomaly depicted in Figure 8d against the HDT message is detected, the following is known:

- An inconsistency is identified by a physical-based specification concerning the heading information in HDT messages; this is due to steps 6 and 3. In step 3 the heading field was designated for identifying relevant anomalies, attacks, and detection methods. In step 6 the detection methods for the heading field were proposed.
- The anomalous messages are arriving from the gyro compass; this is known through the TalkerID identified in step 2.
- This might indicate a data field evolution anomaly; identified in step 4.
- This might be a result of a MoV attack; the relationship between the anomaly and the attack that possibly causes it is identified in step 5 (see Table 5).
- This could impact the Route Monitoring, Navigation control data and Navigational status and data display functions in the INS which might cause safety, financial and reputation loss and environmental damage; the relationships between the targeted message and the relevant functions are identified in step 1 (see Table 2).

*6.4. Identified Limitations*

The experiments have highlighted some possible limitations in the different anomaly detection approaches. A summary of the identified limitations is provided below:

- False positives: weakly configured specifications can generate false positive alerts. For instance, a system specification for one simulated ship is a range for RPM change of 100 between two consecutive RPM messages. Using the same specification in another ship with a different RPM change range would generate a false positive. Therefore, it is highly encouraged to fine-tune the system specifications for each target system. Another aspect to consider is the sensor error or noise. A noise in the sensor might invoke an anomaly and a false indication of malicious behavior. An instance of this issue has been observed in one of the experiments. A glitch in the simulator caused the speed of the vessel to abnormally change due to low water depth. An anomaly in the data is observed which is not caused by malicious behavior. These issues need to be considered during the development of the anomaly detection system.
- False negatives: malicious behavior operating within the thresholds defined in the specifications will not be detected but still can cause an impact. For instance, reducing the RPM value by 50 while the change threshold is 100 will not generate an alert, but, the speed value will appear less than what is expected, this can cause a speed increase command which in turn can cause an issue in safe navigation.

  Moreover, other limitations in our analysis are mentioned hereafter:

- The number of analyzed messages is limited to those supported by the available simulator. Still, four message types, namely, GLL, RMC, GGA, ZDA are among the top 10 NMEA messages observed on the internet during a scan in Shodan [50]. Future work should focus on more message types.
- Our analysis included only NMEA0183. Other protocols are being integrated into the maritime systems including NMEA2000 [6] and OneNet [51]. Yet, NMEA0183 is still utilized in the maritime industry as observed in the literature and the internet-wide Shodan scan [50].
- Our analysis only considers attacks with objectives to impact navigational tasks by denying and manipulating the view that is rendered using the NMEA messages. Other attack techniques that can cause anomalies can be investigated in future work.
- We utilized the ATT&CK framework for the threat modeling. Using other threat modeling techniques might identify other attacks. Still, our considered attacks are in line with the attacks discussed in the literature.

- We utilized certain categories of anomalies during our analysis based on the observed anomalies in the literature. Other anomalies can exist. If new anomalies are identified in the future, this would require another iteration of the analysis process to consider relevant messages, fields, attacks, and detection methods.

## 7. Deployment Options

Lokman et al. [31] discussed the observed deployment strategies of IDS for CAN bus. It has been observed that CAN bus IDSs are placed either in the central or end nodes or within the CAN network. NMEA IDSs are expected to have the same options. However, the choice of placement is sensitive to the use case. Still, we discuss here the possible placement of NMEA IDSs in marine systems. Two main categories of IDS are observed in the literature, namely Host-based IDS (HIDS) and Network-based IDS (NIDS). Jacq et al. [52] have discussed the concept of situational awareness in naval systems and indicated the challenge in the application of HIDS due to possible warranty disruptions. On the other hand, NIDS is a more suitable deployment option, as it can be added to the networks for monitoring NMEA traffic and detecting anomalies. However, we argue that if attackers are able to target the ship network and successfully carry the attacks presented in this paper, NIDS might also be targeted using similar attack techniques, to avoid detection. Therefore, we argue that the optimal implementation can be achieved through the integration of the anomaly detection solution within the NMEA receiver node, as part of the software that consumes NMEA messages. Still, considering that this solution would allow real-time anomaly detection, a performance evaluation is crucial to validate that the solution does not hinder the navigation functions. Future work can target a proof-of-concept implementation through the development of an NMEA IDS and integrate it with the OpenCPN software.

## 8. Conclusions

The ongoing digitization in maritime is leading to new operational modes. The shipping operations are being gradually transferred to remote shore locations, relying on sensor data transmitted from the vessels. This mode of operation makes the shipping operations susceptible to a wide range of cyber-attacks including manipulation and denial of view, which subsequently hinders safe navigation. This paper targeted the detection of anomalies in NMEA messages caused by malicious actors. NMEA messages carry information that is crucial for several navigational functions, such as collision avoidance. The consequences of targeting them in cyber attacks could cause safety, operational and financial consequences.

Initially, a systematic analysis of NMEA messages was proposed. The analysis aims to identify anomalies in NMEA messages, their cause, and possible detection methods. Afterwards, several of the identified anomalies were invoked using some of the identified attacks in a testing environment against a group of simulated NMEA messages. Then the identified detection methods were evaluated.

Our analysis suggests relevant NMEA messages and fields which have demonstrated utility for detecting inconsistencies. Moreover, the systematic analysis provides a multi-tier overview of the risks associated with the detected anomalies. When an anomaly is detected at the technical tier, technical-level information can be induced, such as source device, candidate attack technique, and relevant messages for investigation. Furthermore, risk information is utilized at the operational or mission tier regarding the possible risk of the detected anomaly against the relevant functions. The employed attack techniques reflect several possible threat actors with varying degrees of complexity; this constitutes a good coverage of possible threats against the NMEA messages within scope. Other attack techniques to achieve other objectives than impacting the navigational functions will be considered in future work.

Specification-based and frequency-based detection has been demonstrated to provide detection capability using different approaches. Protocol specifications can provide a confident indication of conformity issues in the NMEA messages. Furthermore, system specifications can provide a confident indication of system-specific anomalies related to

some values and events that are not expected in a specific system. However, the efficiency of this approach relies on the strength of the defined specifications for each host system. Advanced attacks can still avoid detection. Additionally, physical specifications can provide indicators of anomalies, yet they require careful development of the specifications. We have demonstrated the utility of identifying relationships among some NMEA messages and their fields in the identification of data field evolution anomalies. The relationships can provide strong indications of anomalous patterns. Still, a comprehensive analysis of all the identified relationships is required to generalize the findings. Moreover, frequency-based detection can provide a strong indication of over and under reporting anomalies. However, advanced attackers can avoid detection by maintaining a transmission rate that is consistent with normal traffic.

Future work can utilize the results of our analysis to develop, implement, and evaluate an NMEA anomaly detection solution that is suitable for deployment onboard vessels. Furthermore, another direction could utilize a variation of our testing environment by including other NMEA simulators or a physical NMEA source (e.g., GPS or AIS); this would include other NMEA messages and fields. Additionally, this paper targeted attack techniques that can cause an impact on navigational functions. Future work can explore different attack techniques that do not aim to impact the navigational functions and can still invoke anomalies such as ex-filtration or command and control. Moreover, we have identified a possible application of machine learning techniques in NMEA anomaly detection. Yet, the limited amount of data was a challenge that can be targeted in future work. Finally, another direction for future work can be the extraction of physical and system specifications from a large amount of NMEA messages recorded in several systems, to improve the specification rules.

**Author Contributions:** Conceptualization, A.A., V.G. and S.K.; Data curation, A.A.; Formal analysis, A.A. and A.O.; Investigation, A.A.; Methodology, A.A.; Software, A.A.; Supervision, V.G. and S.K.; Validation, A.A., A.O. and V.G.; Writing—original draft, A.A., A.O.; Writing—review & editing, A.A., V.G. and S.K. All authors have read and agreed to the published version of the manuscript.

**Funding:** This work was funded by (a) the NTNU Digital transformation project Autoferry; (b) the Research Council of Norway through the Maritime Cyber Resilience (MarCy) project, Project no. 295077; and (c) the Research Council of Norway through the SFI "Norwegian Centre for Cybersecurity in Critical Sectors (NORCICS)", Project No.310105

**Institutional Review Board Statement:** Not applicable.

**Informed Consent Statement:** Not applicable.

**Data Availability Statement:** The data generated during the data generation process discussed in Section 5.1 can be shared upon request.

**Conflicts of Interest:** The authors declare no conflict of interest.

## Abbreviations

The following abbreviations are used in this manuscript:

| | |
|---|---|
| NMEA | National Marine Electronics Association |
| APS | Autonomous Passenger Ship |
| INS | Integrated Bridge System |
| IMO | International Maritime Organization |
| AIS | Automatic Identification System |
| ECDIS | Electronic Chart Display and Information System |
| GPS | Global Positioning System |
| CAN | Controller Area Network |
| MoV | Manipulation of View |
| DoV | Denial of View |

ARG  Attack Rules File
NMF  NMEA Messages File
UTC  Coordinated Universal Time
CSV  Comma Separated Valu
TR   Transmission Rate
SOG  Speed Over Ground
COG  Course Over Ground
DTM  Datum Reference Message
GLL   Geographic position Message
RMC  Recommended Minimum specific GNSS data Massage
RPM  Revolutions Per Minute Message
TTM  Tracked Target Message
GGA  Global positioning system (GPS) fix data Massage
HDT  Heading true Massage
ROT   Rate of Turn Massage
RSA   Rudder Sensor Angle Massage
ZDA  Time and Date Massage

## Appendix A. NMEA Messages and Their Fields

The investigated NMEA messages and their fields in steps 2 and 3 are depicted in Figure A1. For more details please refer to the IEC 61162-1 standard.

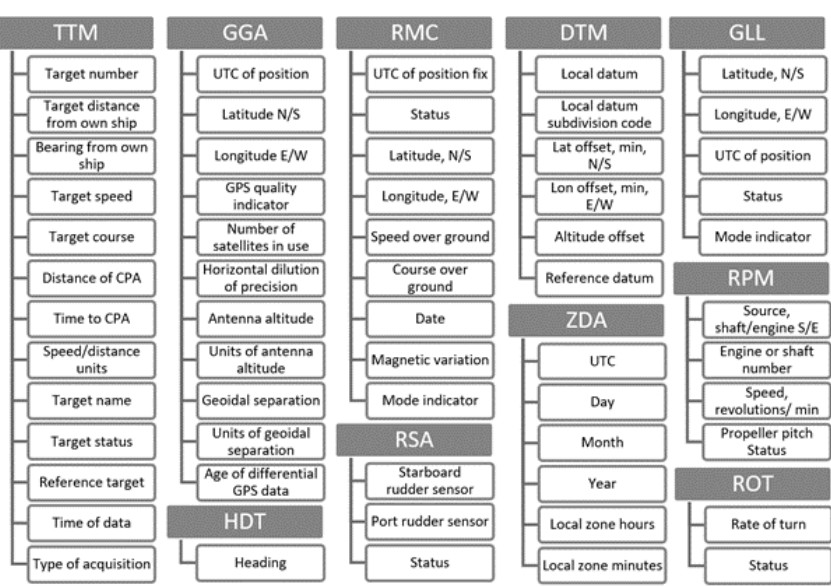

**Figure A1.** A list of the analyzed messages and their fields.

## Appendix B. Interaction of NMEA Messages

In Table A1, the relevance among the analyzed NMEA messages is presented. This is an outcome of step 3.

**Table A1.** Relevant fields across the selected NMEA messages.

| Msg | Fields | Related Msg | Related Fields |
|---|---|---|---|
| DTM | Local datum | GGA, GLL, RMC | Longitude, E/W, Latitude, N/S |
| GGA | Longitude, E/W, Latitude, N/S | RMC | COG, Longitude, E/W, Latitude, N/S |
| | | ROT | Rate of turn |
| | | HDT | Heading |
| | | RSA | Rudder angle |
| | | DTM | Local datum |
| | | GLL | Longitude, E/W, Latitude, N/S |
| GLL | Longitude, E/W, Latitude, N/S | GGA | Longitude, E/W, Latitude, N/S |
| | | RSA | Rudder angle |
| | | DTM | Local datum |
| | | RMC | COG, Longitude, E/W, Latitude, N/S |
| | | ROT | Rate of turn |
| | | HDT | Heading |
| HDT | Heading | RSA | Rudder angle |
| | | RMC | COG |
| | | ROT | Rate of turn |
| | | GLL, GGA | Longitude, E/W, Latitude, N/S |
| | | TTM | Bearing Time to CPA, Distance of CPA |
| RMC | COG, Longitude, E/W, Latitude, N/S | RSA | Rudder angle |
| | Longitude, E/W, Latitude, N/S | DTM | Local datum |
| | COG | ROT | Rate of turn |
| | | HDT | Heading |
| | COG Longitude, E/W, Latitude, N/S | GLL, GGA | Longitude, E/W, Latitude, N/S |
| | SOG | RPM | RPM (if FPP) Propeller pitch (if CPP) |
| ROT | Rate of turn | RMC | COG |
| | | HDT | Heading |
| | | RSA | Rudder angle |
| | | GLL, GGA | Longitude, E/W, Latitude, N/S |
| | | TTM | Bearing Time to CPA, Distance of CPA |

FPP: Fixed Pitch Propeller, CPP: Controllable Pitch Propeller, CPA: Closest Point of Approach.

## Appendix C. The Identified Anomalies

The identified anomalies in the investigated messages and their fields in step 4 (Section 3.4) are presented in Table A2. These anomalies are relevant to the NMEA0183 protocol and can therefore be utilized in use cases other than the INS and APS.

**Table A2.** The identified anomalies in the NMEA messages within scope.

| Anomaly | Message | Field | Anomaly | Message | Field |
|---|---|---|---|---|---|
| Sudden unexpected change | RPM | RPM in FPP | Unexpected value | TTM | Target Speed |
| | | Propeller pitch in CPP | | | Time until CPA |
| | GGA | Location (long. & lat.) | | DTM | All Fields |
| | | UTC | | ZDA | Local zone |
| | | GPS Quality Indicator | | | Local zone minutes |
| | | Age of differential GPS data | | RMC | SOG |
| | GLL | Location (long. & lat.) | | | Magnetic Variation |
| | | UTC | | GGA | GPS Quality Indicator |
| | | Status | | | Number of satellites |
| | ZDA | Time (UTC, day, month, year, zone, and zone minutes) | | | Horizontal Dilution |
| | HDT | Heading | | | Antenna Altitude |
| | | T = True | | | Geoidal separation |
| | ROT | Rate Of Turn | | | Age of differential GPS data |
| | | Status | | ROT | Rate Of Turn |
| | RSA | Starboard rudder sensor | Incorrect value | GGA | UTC |
| | | Port rudder sensor | | | UTC |
| | RMC | Location (long. & lat.) | | ZDA | Day |
| | | Status | | | Month |
| | | SOG | | | Year |
| | | COG | | GLL | UTC |
| | | Date | | RMC | UTC |
| | | UTC | | | Date |
| | | Nav status | | TTM | UTC |
| | TTM | Distance, bearing, speed, course, distance of CPA, time until CPA, units, and target status | Data field evolution | All Messages | All fields |
| | | UTC | Conformity issue | | |
| | | Type | Under Reporting | | |
| | DTM | All fields | Over Reporting | | |
| Nonexistent value | TTM | Target Distance | | | |
| | | Bearing from own ship | | | |
| | | T or R | | | |
| | | Distance of CPoA | | | |
| | | Speed / distance units | | | |
| | DTM | Local datum code | | | |
| | | Local datum subcode | | | |
| | | Datum name | | | |
| | RPM | Source | | | |
| | | Source number | | | |
| | | RPM (i.e., speed) | | | |
| | | Propeller pitch | | | |
| | RSA | Starboard rudder sensor | | | |
| | | Port rudder sensor | | | |
| | HDT | Heading | | | |
| | | T = True | | | |

## Appendix D. Data Generation Experiments

A summary of the conducted experiments during the data generation process discussed in Section 5.1 is presented in Table A3. The experiments resulted in generating normal and attack data for consequent anomaly detection steps. The table depicts the

targeted anomalies for invocation, the conducted attack techniques, the targeted messages, and a brief description of the attack.

**Table A3.** Summary of conducted experiments during data generation.

| # | Attack Type(s) | Anomaly(s) | Message(s) | Description |
|---|---|---|---|---|
| 1 | MoV: Fixed | Sudden unexpected change | RPM | Fixed RPM to (60) and propeller pitch to (10) and Fixing True field in HDT to (R) |
| | | | HDT | |
| | MoV: Context | | GGA, GLL, RMC | Changing position: Decreasing latitude and longitude by 45 degrees (4500) and minutes by (30.000) |
| | | Nonexistent value | RPM, RSA | Changing source of RPM to Engine and number of source to 4. Streaming two sensor values for RSA starboard and port |
| | Mov: Fixed | Unexpected value | GGA, ROT | Fixed HDoP to (50.0) and increase ROT by 100 |
| | MoV: Context | | GGA, GLL | Go back in time 1 day by changing the UTC field |
| | | | TTM, DTM | Increase target distance by app 460 meters or 0.25 nautical miles (0.25), distance of CPA by (0.25), time until CPA by (1.0) and Modify Datum specs for default datum |
| | | Data field evolution | RPM | Increment RPM by (10) |
| 2 | MoV: Context | Data field evolution | RMC | Decreasing SOG by (10.0) and Increasing COG by (45.0) |
| | | | ROT | Increasing ROT by 10 |
| | | | HDT | Increasing heading by 5 |
| | | | TTM | Losing/recovering targets fluctuation |
| | DoV | Under reporting | ALL | Drop messages |
| 3 | Mov: Fixed | Conformity issue | RPM, RSA, DTM, GGA | RPM and RSA status fields to M. DTM offsets to string. GGA number of sat. to x |
| | MoV: Confusion | Over reporting | All | Confusiona attack |
| | Mov: Replay | Several | All | Replay attack |

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
