# Peer review of "Navigation Data Anomaly Analysis and Detection"

_information, doi:10.3390/info13030104_

Round 1

Reviewer 1 Report

The peer-reviewed work investigates and analyzes different types of anomalies in navigation information.

Remarks:

  1. The article has a lot of abbreviations, which complicates the perception and understanding of the material. Moreover, there is a repetition of the interpretation of abbreviations (for example, INS, APS in Abstract and Introduction, point 4.1, Abbreviations), some abbreviations are not described (for example, DTM, RMC, etc.). Not all abbreviations are included in Abbreviations.
  2. There is no clear (mathematical) definition of the anomaly. Mathematical methods for detecting anomalies are not given. Everything is given in descriptive, situational form.
  3. The method of detecting anomalies was chosen experimentally (point 4.6), there are no comparisons with other similar methods.
  4. It is not clear how the authors mathematically evaluate the measure “large differences between consecutive messages” (point 6.1.4.)
  5. The situation of the influence of data noise on the detection of anomalies has not been studied.
  6. There are a lot of figures and tables in the work, some of which do not contain important information, but only complicate the understanding of the ideas and methods proposed by the authors.

Author Response

See the attached document for our responses. Modifications to the original text appear in red font.

Reviewer 2 Report

The problem of Navigation Data Anomaly Analysis and Detection is very actual. The general problem of system security is taken more and more seriously as the number and types of attacks are constantly increasing. 

This paper presents a systematic and multidimensional anomaly analysis of NMEA messages and analysis of possible anomalies in NMEA messages and their cause-and- effect relationship with a range of cyber-attacks. 

The article is interesting, describing in detail the analysis of typical threats in in the maritime industry. Very interesting problem and interesting presentation. 

A rich review of literature and other solutions. Authors wanted to improve the security of NMEA messages by identifying and proposing relevant approaches for the treatment and monitoring of the risks associated with them. 

In paper authors proposed a method for creating synthetic datasets with both normal and maliciously tampered with NMEA messages, and original implementation  and software package to create such experimental datasets. 

Specification-based and frequency-based detection has been demonstrated to provide detection capability using different approaches.

The research topic is current and should be further developed. No errors related to technical terminology were found. The topic was presented in a comprehensive way. They describe some of the current and future challenges for research. The paper is correct, quite detailed and presented in a proper way. The paper is written in clear and understandable language.

A few issues can be more clarified:

Why authors use only two cases to evaluate the performance of anomaly detection?

How data encryption process can be affect to exploited to hide malicious activities/corrupt the data into normal message transfer.

Do the authors have a database of signatures, anomalies, etc. What size of this data is appropriate to correctly diagnose the threat. 

Is the number of message types finite? Can the number of defined anomaly types be extended? How it affect the proposed solution.

Are there other apps for threat modeling due to its comprehensive nature and suitable level of abstraction. If so, it is worth comparing their operation.

How is the real effectiveness of threat detection - percentage value.

Did the authors carry out tests on a real system (or an isolated system) taking into account historical data on attacks? Has detection performance been assessed?

Author Response

(The authors gave the same response as above.)

Reviewer 3 Report

This is an interesting work. It is presented in comprehensive details.

One thing that would be helpful is a discussion about the processing requirements. Would a requirement for a high processing be a hindrance to the application. Is the proposed analysis intended for a real time attack detection?

Author Response

(The authors gave the same response as above.)

Round 2

Reviewer 1 Report

Remark 1. not fully corrected. For example, INS, APS in Abstract and Introduction remained. There are abbreviations that are entered once and are not used anywhere else, such as MASS. This needs to be corrected.   The authors responded correctly to other remarks by making appropriate changes to the text. The work has only improved as a result

Author Response

  1. Regarding abbreviation repetition, we are following the guidelines provided by MDPI (https://www.mdpi.com/authors/layout#_bookmark14). Section 3.5 states "Note that the abstract, main text and figure/table/scheme captions are treated separately for abbreviations. This means that you need to define the abbreviation the first time you use it in each part". 
  2. Some abbreviations are only mentioned once but mentioning them was deemed necessary for the maritime audience. Please refer to MDPI's author guidelines which state that the appearance of the abbreviation depends on the intended audience. This includes SOLAS, GMDSS, and MASS.
    •  SOLAS is a known maritime convention that is referred to using the abbreviation. 
    • MASS refers to an official class of vessels that is within the scope of this paper. 
    • GMDSS refers to a known system in maritime that is referred to using the abbreviation. 
  3. Some abbreviations were removed, namely, OCSVM, VDR and OOW.